# Anomaly Detection Neural Network with Dual Auto-Encoders GAN and Its Industrial Inspection Applications

**DOI:** 10.3390/s20123336

**Published:** 2020-06-12

**Authors:** Ta-Wei Tang, Wei-Han Kuo, Jauh-Hsiang Lan, Chien-Fang Ding, Hakiem Hsu, Hong-Tsu Young

**Affiliations:** 1Department of Mechanical Engineering, National Taiwan University, Taipei 10617, Taiwan; f06522717@ntu.edu.tw (T.-W.T.); r07522716@ntu.edu.tw (W.-H.K.); r07522749@ntu.edu.tw (J.-H.L.); hyoung@ntu.edu.tw (H.-T.Y.); 2Taiwan Instrument Research Institute, National Applied Research Laboratories, Hsinchu 30076, Taiwan; 33DFAMILY Technology Co. Ltd, New Taipei 23674, Taiwan; hakiem@3dfamily.com

**Keywords:** automated optical inspection (AOI), anomaly detection (AD), defect detection, generative adversarial network (GAN), dual auto-encoder generative adversarial network (DAGAN)

## Abstract

Recently, researchers have been studying methods to introduce deep learning into automated optical inspection (AOI) systems to reduce labor costs. However, the integration of deep learning in the industry may encounter major challenges such as sample imbalance (defective products that only account for a small proportion). Therefore, in this study, an anomaly detection neural network, dual auto-encoder generative adversarial network (DAGAN), was developed to solve the problem of sample imbalance. With skip-connection and dual auto-encoder architecture, the proposed method exhibited excellent image reconstruction ability and training stability. Three datasets, namely public industrial detection training set, MVTec AD, with mobile phone screen glass and wood defect detection datasets, were used to verify the inspection ability of DAGAN. In addition, training with a limited amount of data was proposed to verify its detection ability. The results demonstrated that the areas under the curve (AUCs) of DAGAN were better than previous generative adversarial network-based anomaly detection models in 13 out of 17 categories in these datasets, especially in categories with high variability or noise. The maximum AUC improvement was 0.250 (toothbrush). Moreover, the proposed method exhibited better detection ability than the U-Net auto-encoder, which indicates the function of discriminator in this application. Furthermore, the proposed method had a high level of AUCs when using only a small amount of training data. DAGAN can significantly reduce the time and cost of collecting and labeling data when it is applied to industrial detection.

## 1. Introduction

To solve the problem of manual inspection, automated optical inspection (AOI) that uses image processing algorithms for industrial inspection has been developed [1,2,3]. Furthermore, the automatic detection system has been applied in computer diagnosis tasks, such as monitoring respiration symptoms in body area networks [4]. However, AOI is limited as it can only perform inspection tasks with a simple background and single defect type. Recently, researchers have started to apply convolutional neural networks (CNN) to image recognition, and successively proposed classic CNN architectures such as VGG [5], Inception [6,7,8], ResNet [9], and DenseNet [10]. CNNs have a greater classification ability compared with traditional image processing algorithms. A growing number of studies have begun to use CNNs for defect detection tasks, such as for inspecting cement surfaces [11], industrial products [12], catenary split pins in high-speed railways [13], and cracks due to its outstanding performance [14]. Accordingly, the CNN has been introduced into the industry to improve the capabilities of AOI. However, the serious problem of imbalanced samples arises when it is applied, where there are more normal samples than anomaly samples. Although a variety of data augmentation methods have been proposed to address this issue [15,16,17], CNNs are still limited in industrial inspection situations. Consequently, anomaly detection has been developed to account for this issue. One of the promising anomaly detection algorithms utilizes the generative adversarial network (GAN) to produce images with a similar probability distribution of the training data for anomaly detection. This GAN-based anomaly detection technology has received increased attention, and many networks, such as AnoGAN [18], GANomaly [19], and Skip-GANomaly [20], have been proposed sequentially. The pipelines of these three networks are shown in Figure 1. Furthermore, the GAN-based architecture has been applied to detection for time series data [21,22] and facial expression synthesis [23] and showed impressive ability.

In recent years, researchers have improved the image reconstruction ability of GAN [24,25,26] using CNN and batch normalization [24], Wasserstein loss [25], and dual auto-encoder architecture [26]. This study proposed a GAN-based anomaly detection neural network with dual auto-encoders (DAGAN) to enhance GAN-based anomaly detection in the industry. Furthermore, a series of studies on DAGAN’s industrial detection capabilities were conducted:Training and verification of DAGAN using the public industrial inspection dataset, MVTec AD, and comparing it with previous GAN-based anomaly detection networks.Verification of DAGAN’s detection ability in an actual production line with two datasets (surface glass of mobile phone and wood defect detection datasets).Verification of DAGAN’s inspection capability with less training data.

## 2. Related Works

### 2.1. Generative Adversarial Network (GAN)

GAN [27] is an unsupervised learning neural network that learns to generate images with a probability distribution similar to that of the training data. The network uses the game theory to design the loss function of the neural network, where the generator and discriminator compete, for training.

### 2.2. Boundary Equilibrium Generative Adversarial Network (BEGAN)

BEGAN [26] is a GAN model released by Google. By designing both the generator and discriminator as an auto-encoder, BEGAN ensures that the training is more stable and easier to converge to the expected balance point. Its image reconstruction ability is better than that of GAN, and it does not have to consider model collapse and training imbalance.

### 2.3. AnoGAN

AnoGAN [18] is the first attempt to use GAN for anomaly detection. Its main objective is to use normal samples to train GAN, which will generate a fake image with a probability distribution similar to that of the normal sample. By defining the threshold of residual score between the image to be tested and fake image, the network can recognize anomaly samples. However, it requires a significant amount of computing resources.

### 2.4. GANomaly

Samet Akcay et al. [19] developed GANomaly. Unlike AnoGAN, GANomaly does not need to minimize the residual score between the detection image and generated fake image through iteration, but directly creates the fake image after the image is imported by the encode–decode generator, which greatly reduces the computing resources and improves the anomaly detection ability of GANomaly. However, the image reconstruction ability of GANomaly is still not stable in all tasks.

### 2.5. Skip-GANomaly

Samet Akcay et al. [20] proposed an improved model of GANomaly, Skip-GANomaly. Inspired by U-Net [28], the architecture of skip-connection was added to Skip-GANomaly, which exhibits an outstanding ability to reconstruct images. The performance of Skip-GANomaly is more stable than that of AnoGAN and GANomaly. However, Skip-GANomaly does not perform well in all dataset categories, which might be caused by model collapse during the training process.

The advantages and limitations of AnoGAN, GANomaly, and Skip-GANomaly are presented in Table 1. Inspired by Skip-GANomaly, the proposed method, DAGAN, has been designed with a highly stable and excellent network architecture of GAN-based anomaly detection to overcome the limitation of the previous works.

## 3. Proposed Method

### 3.1. Pipeline

The pipeline of DAGAN, as shown in Figure 2, comprises a generator and discriminator. Inspired by Skip-GANomaly [20] and U-Net [28], generator G (.) is designed as an auto-encoder with skip-connection architecture. It can generate a fake image, x’, with almost the same probability distribution as that of the input image, x. The skip-connection architecture provides DAGAN with an excellent reconstruction ability. Conversely, DAGAN’s discriminator D (.) is inspired by BEGAN. This discriminator is used to receive the fake image, x’. D (.) can identify the difference between the image, x, and fake image, x’. The dual auto-encoder architecture ensures that DAGAN training is more stable and easier to converge to the best balance point. In the training process, only the normal samples are input, which provides the generator with better reconstruction ability for a normal sample than for an anomaly sample. Hence, one can identify normal and anomaly samples using a proper residual score, which is defined to represent the residual between image x to be tested and fake image x’. The proposed method maintains the advantages of Skip-GANomaly and BEGAN in architecture design and has strong image reconstruction ability and training stability.

### 3.2. Training Objective

To achieve the goal of anomaly detection, this study has referenced and improved the loss functions of Skip-GANormaly and BEGAN. The loss functions are presented as follows:Adversarial loss: To provide the generator with the best image reconstruction ability, the adversarial loss function is referred. This loss function, as shown in Equation (Equation 1), will reduce the difference between input image x and generated fake image G(x) as much as possible when training generator G(.), whereas discriminator D(.) will distinguish the original input image, x, and fake image, x, generated by generator G(.) as much as possible. The goal is to minimize the adverse loss of generator G(.) and maximize the adverse loss of discriminator D(.). The adversarial loss can be expressed as:
(1)$$L_{adv}=E_{x∼p_{x}}[||D\left(x\right)-{D\left(G\left(x\right)\right)||}_{2}]$$Contextual loss of generator: To provide generator G(.) with better image reconstruction ability, the proposed method uses a contextualized loss function to represent the difference between x and G(x) pixels. It is defined as the L2 distance between the input graph, x, and generated fake image, G(x). This ensures that the fake image is consistent with the input image as much as possible. The equation of contextual loss of generator is defined as:
(2)$$L_{Gcon}=E_{x∼p_{x}}[||x-{G\left(x\right)||}_{2}]$$Contextual loss of discriminator: To converge to the best balance point shortly during training, a contextual loss of discriminator is set. This loss is used to represent the L2 distance between image x and image D(x) formed by the discriminator. This ensures that the original image and image generated by the discriminator is consistent with the input image as much as possible. A contextual loss of discriminator is defined as:
(3)$$L_{Dcon}=E_{x∼p_{x}}[||x-{D\left(x\right)||}_{2}]$$

In the training process, DAGAN can be trained by the weighted summation of the above three loss functions. The definition of the weighted summation loss function is as follows:(4)$$L=\lambda_{adv}L_{adv}+\lambda_{Gcon}L_{Gcon}+\lambda_{Dcon}L_{Dcon}$$
where $\lambda_{adv}$, $\lambda_{Gcon}$ and $\lambda_{Dcon}$ are the weights of three loss functions.

### 3.3. Detection Process

To perform the task of anomaly detection, it is necessary to design the detection process, as shown in Figure 3. First, image x, which is to be tested, is input into generator G(.). After the generator generates the fake image, G(x), the residual score, R(x,G(x)), between x and G(x) is calculated through the residual score calculator, which is defined as
(5)$$R\left(x,G\left(x\right)\right)=||x-{G\left(x\right)||}_{2}$$

The residual score, R(x,G(x)), of the normal sample will be lower because only normal samples are trained. Through the calculation of the residual score, the residual score, R(x,G(x)), of the entire dataset is linearized to the range of 0∼1 to facilitate the subsequent setting of thresholds and analysis. By adjusting different thresholds θ, when the residual score of the test image is greater than or equal to the test threshold, i.e., R(x,G(x))≥θ, then the product to be tested is an anomaly product.

## 4. Experimental Setup

### 4.1. Datasets

To ensure that the proposed method has good detection ability in industrial inspection, three datasets were used to train and verify DAGAN. The datasets are described below:

#### 4.1.1. MVTec AD

The MVTec AD dataset [29] was collected by the MVTec software GmbH team. It contains 15 common industrial inspection categories: five of them are texture categories and ten are object categories. The data are shown in Figure 4. This dataset is commonly used for validation of industrial detection deep learning models [30,31,32]. In this dataset, there are 3629 training images and 1725 verification images, and the resolution of the image is between 700 × 700 and 1024 × 1024.

#### 4.1.2. Production Line Mobile Phone Screen Glass Dataset

In this study, a line scan camera was used to take the images of mobile phone screen glass pieces, and the images were divided into normal and anomaly samples. The normal and anomaly images in the dataset are shown in Figure 5.

Notably, in the industrial inspection, there was dust adsorption on the mobile phone screen glass. However, the dust can be removed only by wiping, and thus, it poses a challenge in the mobile phone screen glass detection. In this dataset, 200 pieces of mobile phone cover glass were scanned using the camera. There were 329 training and 54 validation images. The image resolution was 128 × 128.

#### 4.1.3. Production Line Wood Surface Dataset

The wood surface dataset contained images of normal and anomaly wood products that were captured by a line scan camera. This dataset comprised six labels, such as normal products, chalk, holes, black, and knots. The sample images of each category are shown in Figure 6. This dataset contained 3075 training data of normal samples and 740 validation data of normal samples and anomaly images. The resolution of the image was 256 × 256.

### 4.2. Training Detail

To ensure that the training is fast and effective, Adam was used as an optimizer, and the learning rate was set to 0.001. The loss function has been defined in Equation (Equation 4). The weights of the loss function were set to $\lambda_{adv}=1$, $\lambda_{Gcon}=40$, $\lambda_{Dcon}=1$, and the number of training steps was set to 20,000. Furthermore, the detection ability of a U-Net auto-encoder was applied to test the necessity of the discriminator. In this study, the model with the best detection ability in the training process was used to verify the results. The experimental hardware used in this work was an Intel i7-9700k 3.6 GHz CPU (INTEL MICROELECTRONICS ASIA LTD., TAIWAN, Taipei, Taiwan) and anlNvidia RTX 2080ti 11 Gb GPU (GIGABYTE Technology, New Taipei, Taiwan), and keras was used as the deep learning framework for training and verification.

### 4.3. Evaluation

The area under the curve (AUC) of the receiver operating characteristics (ROC) was used to evaluate the performance of detection in this study. The AUC is an effective method to evaluate the detection ability of a binary detection model, which is also widely used as a model evaluation method of deep learning.

## 5. Experiment Results

### 5.1. MVTec AD Dataset

As summarized in Table 2 and Figure 7, the proposed method had the best performance in 9 of the 15 categories of MVTec AD. In addition, in the other 6 categories, while its AUC was not the highest, it was almost the same as the highest value obtained. Notably, the detection ability of the proposed method in the four categories of carpet, hazelnut, tile, and toothbrush was significantly higher than that of AnoGAN, GANomaly, and Skip-GANomaly.

These four detection tasks were similar in that they have a relatively complex variation information. The backgrounds of carpet and tile had an irregular texture, hazelnut was the only category with inconsistent sample orientation in MVTec AD, and there were multiple colors of bristles in the toothbrush. When training these more complex tasks, AnoGAN and GANomaly had difficulty in reconstructing the images. Although Skip-GANomaly had the architecture of skip connection, the complexity of the image might have increased the possibility of mode collapse during its training. However, because the proposed method had a strong image reconstruction ability and was easy to converge to the best balance point in the training process, its advantages were remarkable in the categories with high complexity. Additionally, the AUCs of DAGAN were significantly higher than those of the U-Net were in the four categories, i.e., carpet, metal nut, toothbrush, and transistor. This is because without the discriminator, the only goal of the U-Net auto-encoder is to reconstruct the image perfectly. Consequently, it reconstructed the defect in the image, which decreased its detection ability in these categories. Therefore, the discriminator is necessary in this application. Moreover, Figure 8 shows the heat maps generated by the proposed method after detecting the MVTec AD dataset. Notably, the proposed method can classify the defects, and obtain the location, area, and contour of some defects from the generated heat maps, which is crucial to industrial inspection.

### 5.2. Production Line Mobile Phone Screen Glass and Wood Surface Dataset

Table 3 presents the AUCs of the glass and wood surface defects detected by the proposed method, U-Net auto-encoder, and the other three GAN-based anomaly detection models. As mentioned in Section 5.1, the proposed method had a better detection ability than the other three when the detection image had a more complex variation. Owing to the noise of dust in the good products of the mobile phone screen glass and the variety of wood background textures in the actual production line, both of these datasets require a more solid reconstruction ability to avoid model collapse in the training process. Thus, the AUC was significantly higher when using the proposed method for training and verification. Further, the AUCs of DAGAN were better than those of the U-Net auto-encoder. This indicates that the removal of the discriminator will cause a decline in detection ability, as previously mentioned.

Figure 9 shows the heat maps generated by the proposed method after detecting the mobile phone screen glass and wood surface datasets on the production line. In the same way, the proposed method can also show the residual value of each pixel through the heat maps of these two datasets.

### 5.3. Training with Few Data

In this study, four categories, i.e., bottle, tile, actual production line wood surface, and actual production line glass, which represented the detection of object samples, detection of texture samples, and complex detection items on the production line, respectively, were selected as the test categories for training with few data. A total of $2^{n}\left(0\geq n\geq 7\right)$ images were used for training and to examine the influence of reducing the number of training samples in the proposed method. The AUCs under different n are presented in Table 4 and Figure 10 In the four learning categories, a reduction in the number of training samples has little effect on AUCs, which indicates that the proposed method still has a high reducibility to unfamiliar normal product data. It can significantly reduce the time and cost of collecting and labeling data when it is applied to industrial detection.

## 6. Conclusions

In this study, a GAN-based anomaly detection model, DAGAN, was proposed and discussed. By combining the advantages of Skip-GANomaly and BEGAN, the model showed a great reconstruction ability and stability in the training process. Three datasets, MVTec AD, wood surface defects, and glass surface defects of mobile phones, were employed to train and verify the proposed method and for comparison with the previous GAN-based anomaly detection models. The AUCs of the proposed method were significantly higher than those of the other three GAN-based anomaly detection models were in the categories with high variability or noise. Furthermore, the proposed method exhibited better detection ability than the U-Net auto-encoder. Additionally, this study examined the influence of detection capability with different quantities of training data. In this study, four categories were used, and $2^{n}\left(0\geq n\geq 7\right)$ images were utilized during the training process. The result demonstrated that the proposed method could maintain a high level of AUC even when a small quantity of training data was entered, which indicated that the proposed method has a good ability to reconstruct unfamiliar normal product data.

## Figures and Tables

**Figure 1 sensors-20-03336-f001:**
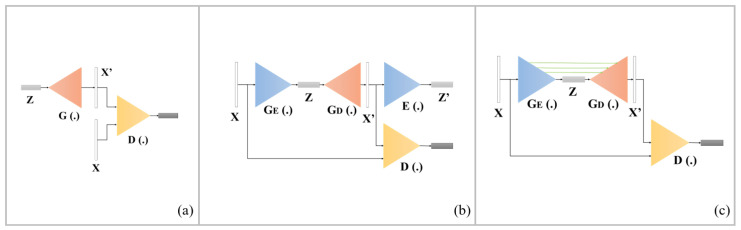
Pipelines of GAN-based anomaly detection networks: (**a**) AnoGAN, (**b**) GANomaly, and (**c**) Skip-GANomaly.

**Figure 2 sensors-20-03336-f002:**
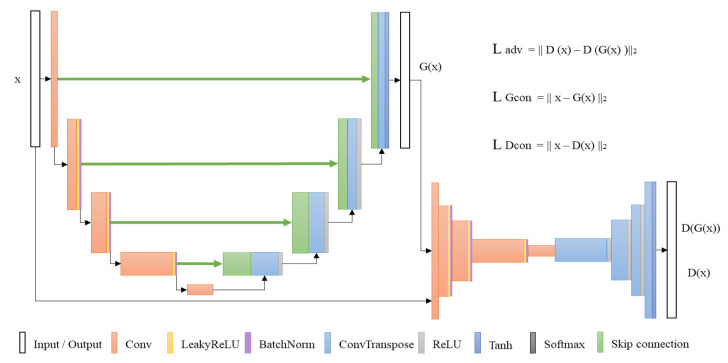
Pipeline of the proposed method (DAGAN).

**Figure 3 sensors-20-03336-f003:**
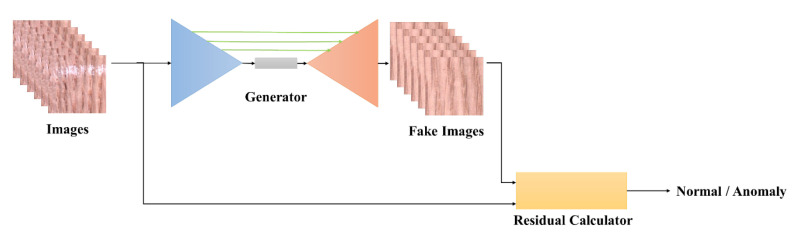
Detection process of the proposed method.

**Figure 4 sensors-20-03336-f004:**
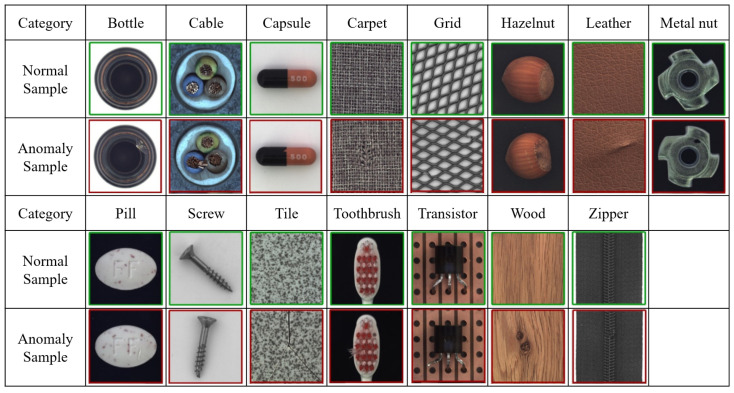
MVTec AD dataset for industrial inspection.

**Figure 5 sensors-20-03336-f005:**
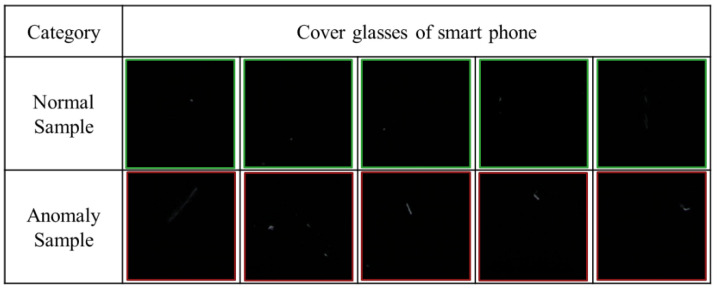
Production line mobile phone screen glass dataset.

**Figure 6 sensors-20-03336-f006:**
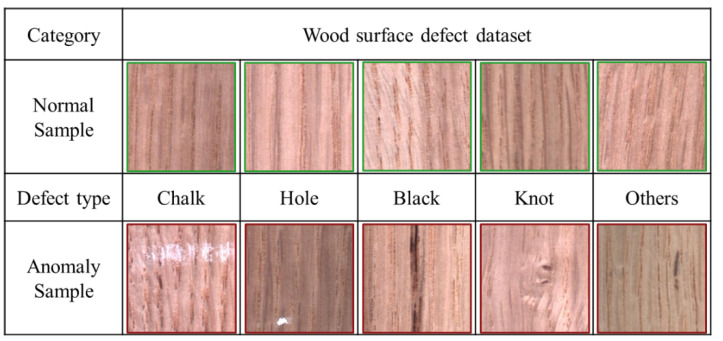
Production line wood surface dataset.

**Figure 7 sensors-20-03336-f007:**
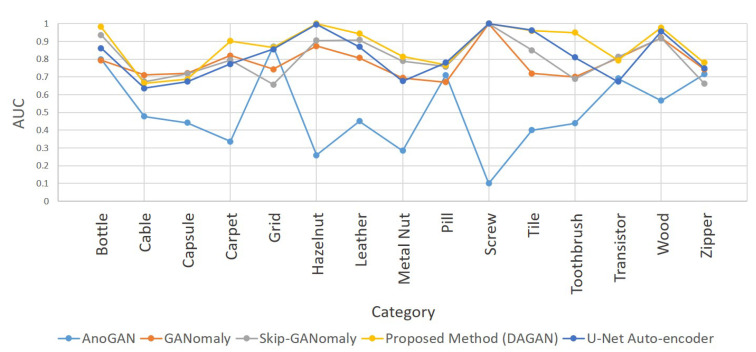
AUC after testing the proposed method (DAGAN) and three other GAN-based anomaly detection models with the MVTec AD dataset.

**Figure 8 sensors-20-03336-f008:**
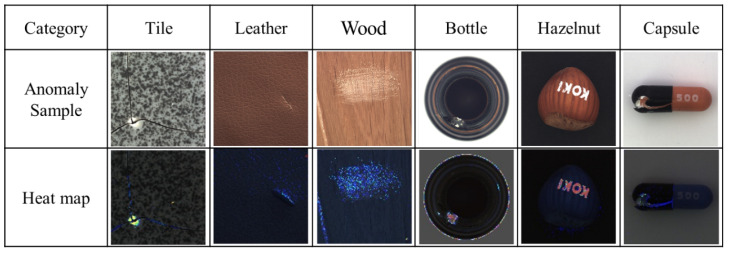
Heat maps of the MVTec AD dataset generated by the proposed method (DAGAN).

**Figure 9 sensors-20-03336-f009:**
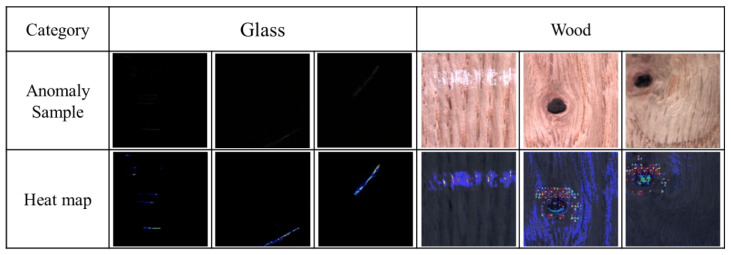
Heat maps of the glass and wood datasets generated by the proposed method (DAGAN).

**Figure 10 sensors-20-03336-f010:**
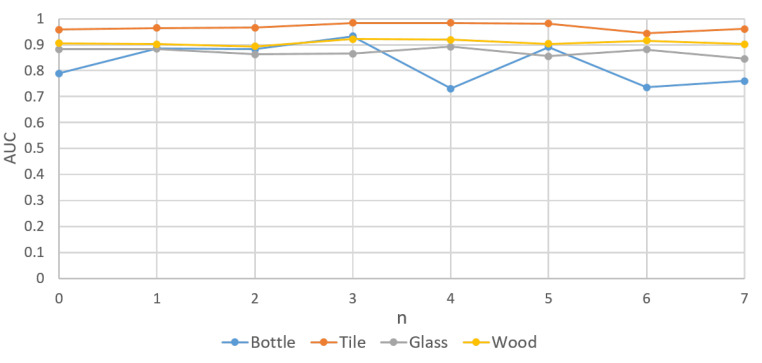
AUCs of training the proposed method (DAGAN) with few data $\left(2^{n},0\geq n\geq 7\right)$.

**Table 1 sensors-20-03336-t001:** Advantages and limitations of AnoGAN, GANomaly, and Skip-GANomaly.

	AnoGAN	GANomaly	Skip-GANomaly
Advantages	Training without anomaly data.	Significant improvement in detection time.	Better ability of image reconstruction.
Limitations	Excessive time to detection.	Cannot reconstruct complex images.	Model collapse during training.

**Table 2 sensors-20-03336-t002:** AUCs of each category in MVTec AD dataset using AnoGAN, GANomaly, Skip-GANomaly, proposed method (DAGAN), and U-Net auto-encoder.

Category	AnoGAN	GANomaly	Skip-GANomaly	DAGAN	U-Net
Bottle	0.800	0.794	0.937	0.983	0.863
Cable	0.477	0.711	0.674	0.665	0.636
Capsule	0.442	0.721	0.718	0.687	0.673
Carpet	0.337	0.821	0.795	0.903	0.774
Grid	0.871	0.743	0.657	0.867	0.857
Hazelnut	0.259	0.874	0.906	1.00	0.996
Leather	0.451	0.808	0.908	0.944	0.870
Metal Nut	0.284	0.694	0.79	0.815	0.676
Pill	0.711	0.671	0.758	0.768	0.781
Screw	0.10	1.00	1.00	1.00	1.00
Tile	0.401	0.72	0.85	0.961	0.964
Toothbrush	0.439	0.700	0.689	0.950	0.811
Transistor	0.692	0.808	0.814	0.794	0.674
Wood	0.567	0.920	0.919	0.979	0.958
Zipper	0.715	0.744	0.663	0.781	0.750

**Table 3 sensors-20-03336-t003:** AUCs of glass and wood datasets generated using AnoGAN, GANomaly, Skip-GANomaly, the proposed method (DAGAN), and U-Net auto-encoder.

Category	AnoGAN	GANomaly	Skip-GANomaly	DAGAN	U-Net
Glass	0.543	0.600	0.618	0.853	0.828
Wood	0.716	0.915	0.797	0.925	0.886

**Table 4 sensors-20-03336-t004:** AUCs of training the proposed method (DAGAN) with few data $\left(2^{n},0\geq n\geq 7\right)$.

*n*	Bottle	Tile	Glass	Wood
0	0.790	0.958	0.882	0.906
1	0.886	0.964	0.883	0.902
2	0.882	0.966	0.863	0.893
3	0.933	0.984	0.865	0.921
4	0.731	0.984	0.892	0.919
5	0.891	0.981	0.856	0.903
6	0.736	0.943	0.881	0.915
7	0.760	0.961	0.846	0.902

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
