# Peer review of "Anomaly Detection Neural Network with Dual Auto-Encoders GAN and Its Industrial Inspection Applications"

_sensors, 2020, doi:10.3390/s20123336_

Round 1
Reviewer 1 Report
The authors of the paper claim to present a new GAN architecture called DAGAN that uses dual autoencoders in both its generator and discriminator and is inspired by SkipGANomaly and BEGAN. This reviewer recommends acceptance for publication once the following major concerns are addressed.
- Firstly, the paper needs a significant revision in English. Incomprehensible language and incoherent narratives are a big impediment to reading this paper.
- The paper is quite long without any significant need or new content. The paper length can be shortened to 1/2 or 2/3 of the current size to approximately 8-10 pages.
- The paper has multiple areas of repetition where the authors say the same thing over and over again and sometimes in the same language. Please correct this and make it terser.
- Lines 6-8: The authors claim that dual-autoencoders GAN is “brand new”. This is markedly false. This architecture is well-known in the ML community. Among many examples, one example is https://ieeexplore.ieee.org/document/8483579
- Line 49-59: The authors use the term “normal data”. To most of the ML community, this would mean data that is normally/Gaussian distributed. Please be careful before using terms that have mathematically specific meanings.
- Line 276: It is clear from the values of the hyperparameters used to control the contribution of the 3 component loss functions to the total loss function that the contextual loss of the generator is crucial since it is weighted as 40 times more important than the other 2 components. The contextual loss of the generator is nothing but the reconstruction loss of the an autoencoder. This reviewer is therefore strongly suspicious that there is NO need of a discriminator or an adversarial loss. In other words, a GAN is NOT required for this problem. Simply training a plain autoencoder should be sufficient since that is what the authors seem to be doing anyway. The authors MUST justify why the neural architecture is a GAN and not simply a plain autoencoder.
- Figure 10 is not required. Plotting table 2 is redundant since there are just 2 categories.
Reviewer 2 Report
The paper has potential, however please address following comments before recommending it for publication.
1 - add quantitative results to abtract part
2 - The abstract part doesnt indicate complete idea about the manuscript, please rewrite it.
3- introduction section is too long. please shorten it.
4 - please make a table presenting comparison of previous work, advantages and limitations in related section
5- please add following state of the art work in reference section as well. Respiration symptoms monitoring in body area networks
6 - there are typos and grammatical mistakes. please remove
Round 2
Reviewer 1 Report
The changes made by the authors are fine. While the paper can be improved further by adding an uncertainty analysis over the results shown in figure 7, this reviewer is fine with accepting the paper in its present form.
Reviewer 2 Report
The author's have addressed all my comments. I would recommend this paper for publication.